

# How do Microtopography Act in the Pedogenic
# Characteristics of Mudstone-derived Soils in Hilly
# Mountainous Regions?
Banglin Luo[1†], Jiangwen Li[1†], Jiahong Tang[2], Chaofu Wei[1,3,4*], Shouqin Zhong[1,3,4*]
[1]College of Resources and Environment/Key Laboratory of Eco-environment in Three Gorges Region
(Ministry of Education), Southwest University, Chongqing 400715, China
[2]District Agro-Tech Extension and Service Center of Shapingba, Chongqing 400000, China
[3]Key Laboratory of Arable Land Conservation (Southwestern China), Ministry of Agriculture,
Chongqing, 400715, China
[4]State Cultivation Base of Eco-agriculture for Southwest Mountainous Land, Southwest University,
Chongqing, 400715, China
[†]These authors contributed equally to this work.
*Correspondence to*: Shouqin Zhong (zhongsq2021@swu.edu.cn) & Chaofu Wei (weicf@swu.edu.cn)
**Abstract.** The topography is a critical factor that determines the characteristics of regional soil
formation. Small-scale topographic changes are defined as microtopographies. As a characteristic
topographic condition in hilly mountainous regions, the redistribution of water and soil materials
caused by microtopography is the main factor affecting the spatial heterogeneity of soil and the
utilization of land resources. In this study, the influence of microtopography on pedogenesis was
investigated using soil samples formed from mudstones with lacustrine facies deposition in the middle
of the Sichuan Basin and developed into hill landforms by erosion of flowing water. Soil profiles were
sampled along the slopes at the summit, shoulder, backslope, footslope, and toeslope positions. The
morphological, physiochemical, and geochemical attributes of profiles were analysed. The results
showed that the soil thickness increased significantly with changes in the soil profile configuration
from the summit to the toeslope, and the profile configuration changed from A-C to A-B-C. The
migration direction of Ca and Na at the summit, backslope, and footslope changed from enrichment to
leaching, whereas that of Al, Fe, and Mg changed from leaching to enrichment. At the summit and
shoulder of the hillslope, weathered materials are transported away by gravity and surface erosion, and





new rocks are often exposed; therefore, the characteristics of soil development is relatively weak.
However, in flat areas such as the footslope and toeslope with sufficient water conditions, the
long-term contact between water, soil, and sediment leads to further chemical weathering, resulting in
highly developed characteristics. Microtopography can affect physicochemical properties, chemical
weathering, and redistribution of water and materials, resulting in differences in pedogenic
characteristics at different slope positions.
**Keywords**. Pedogenic characteristics; Physicochemical property; Mudstone; Microtopography;
Mineralogy
**1 Introduction**

Soil, as an independent natural body, maintains the lives and reproduction of various creatures on the

land surface and also develops and changes under the control of the soil-forming environment.
Evaluating objective and quantitative changes of soil properties caused by soil weathering and the stage
of soil weathering is the first prerequisite for interpreting changes in the soil environment and
improving the soil environment, and the soil development process behind soil resource change cannot
be ignored. Previously, research on topography mainly considered the effect of bioclimates caused by
large topography. The microtopography refers to the concept relative to the macrotopography, which
the amplitudes markedly smaller than the hillslope or basin scales (Thompson et al., 2010). Small-scale
topographic changes are defined as microtopographies (Wang et al., 2022). Microtopography strongly
affects energy, water, and nutrient cycling at local-site scale (Lv et al., 2023). The redistribution of
water and soil material caused by microtopography is the main factor in soil spatial variation and an
obstacle in the utilisation of soil resources (Kokulan et al., 2018), particularly in hilly mountainous
regions. Therefore, it is of great scientific significance to explore changes in soil weathering and
development under microtopographic conditions in hilly mountainous regions.

Pedogenesis refers to the evolution from the profile scale to the regional scale, which includes

significant changes in the soil under physical, chemical, or biological conditions (Leguédois et al.,
2016). Topography is a key factor that controls soil genesis and strongly influences the
physicochemical properties (Baltensweiler et al., 2020). In areas with large terrain slopes, the degree of





physical erosion of rocks is greater and chemical weathering is stronger (Gabet, 2007; Riebe et al.,
2004). Soil surface roughness and properties in different terrains and altitudes affect the development
of water infiltration, runoff, and drainage in different terrain parts (Vidal Vázquez et al., 2005). The
differences in water characteristics, coupled with strong erosion, affect the material distribution during
soil formation, which leads to differences in the degree of soil development in different terrain parts
(Darmody et al., 2005; Luo et al., 2020; Parent et al., 2008; Salako et al., 2007; Veneman et al., 1984).
On the scale of slopes and small watersheds, topography is the prominent factor that affects the
variation in soil properties, and it is also the main structural factor that controls spatial autocorrelation
(Wang et al., 2001). Compared to the equivalent background state without microtopography, the
presence of microtopography increases the proportion of rainfall infiltration (Thompson et al., 2010).
Within the influence of microtopography, the rainfall distribution pattern and operation mode on the
surface are completely different. Multiple studies have suggested that the difference in crop yield at
different topographic positions is mainly due to the change in soil properties caused by soil erosion
with topographic position (Brubaker et al., 1993; Salako et al., 2007). In addition, microtopography can
control a series of geomorphic processes, such as collapse, transportation, and accumulation, to change
the spatial redistribution of light, heat, water, and soil in a small area, thus affecting vegetation growth
(Baltensweiler et al., 2020; Nagamatsu et al., 2003). Moreover, microtopography can accelerate soil
weathering (Phillips et al., 2008). Pal et al., (2003) found that presence of microtopography resulted in
the development of non-alkaline and highly alkaline soils on the upper and lower slopes, respectively,
of the southwestern Indo-Gangetic Plain. Botschek et al., (1996) found that the organic matter content
was highest in the mineral topsoil on the uphill slope and decreased at the foot of the slope.
The mudstone-derived soils in the middle of the Sichuan Basin were mainly formed by flowing
water erosion. The area has predominantly hilly topography formed by flowing water erosion. Heat and
water are redistributed in different locations of topography, combined with severe soil erosion,
resulting in differences in soil particle composition, nutrient content, water holding capacity, drought
resistance, and fertiliser retention. Therefore, the degree of soil development is different. An effective
solution for soil erosion and its associated soil problems in this region is an important issue that we are
currently facing. The aims of the present study were (1) to explain the characteristics of soil



morphology and physiochemistry at different slope positions; (2) to explore the changes in soil
weathering and development under microtopographic conditions; and (3) to clarify the effect of
microtopography on pedogenic characteristics. Ultimately, this work provided a quantitative basis for
regulating the soil-forming process of mudstone soil under artificial conditions.
**2 Materials and methods**
**2.1 Study area**

The study area, Tongnan District and Dazu District, located in the central Sichuan Basin (Fig. 1).

The area has a predominantly hilly topography formed by flowing water erosion and a subtropical
monsoon climate. The soil samples were classified as Cambisols in the WRB classification (IUSS
Working Group WRB, 2022).

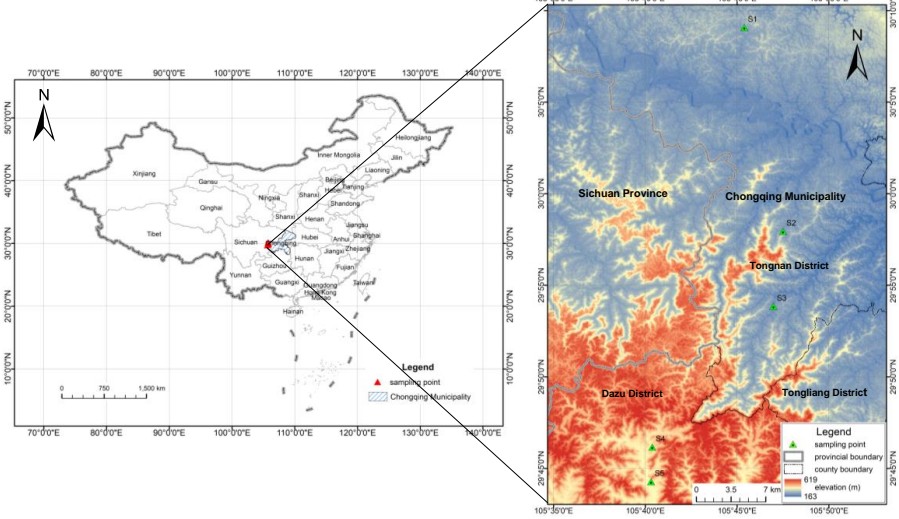


**Figure 1.** Study area. The background DEM dataset (30 m resolution) was downloaded from
Geospatial Data Cloud site, Computer Network Information Center, Chinese Academy of Sciences.
(http://www.gscloud.cn)
**2.2 Soil sampling**

Five natural and typical profiles (according to the microtopography conditions) were selected for

sampling (The information of sampling points was shown in Table 1). According to the 1:200,000



regional geological map of China (https://geocloud.cgs.gov.cn) and the field survey, the parent material
of all the sampling points is purple mudstone of the Upper Jurassic Shaximiao Formation in Mesozoic.
Simultaneously, in order to ensure the similarity, the other soil-forming factors such as climate (mainly
rainfall and temperature) and organisms (grass and crops were the main surface vegetation, and the soil
animals were shown in Table 2) besides terrain of each sampling point were also investigated (Table 1).
Five soil samples with five repetitions were collected at the summit, shoulder, backslope, footslope,
and toeslope positions of the hillslope (Fig. 2). Soil samples were collected from the excavated soil
profile and were sampled from the toeslope to the summit according to the soil genesis characteristics
(divided into horizons A, B, and C). When collecting samples, the GPS locator was used to record the
coordinates, altitude, and slope of the sampling points, and the morphological characteristics of the soil
profiles were recorded according to the *Field Book for Describing and Sampling Soils (version 3.0)*
(Nation Soil Survey Centre et al., 2013) (Table 2). Soil samples were taken to the laboratory for
chemical and physical analyses, totalling 125 samples (approximately 500 g). All the samples were
air-dried. After the removal of visible plant debris, all samples were sieved through a 2 mm sieve for
laboratory analysis.
**Table 1.** The information of sampling points.

| Sampling Point | Coordinates (N/E) | | Elevation (m) | Slope gradient (°) | Average annual rainfall (mm) | Average annual temperature (°C) | Land use | Slope position |
|---|---|---|---|---|---|---|---|---|
| S1-1 | 30°09′04″ | 105°45′31″ | 331 | 10 | 1121.5 | 18.2 | Grassland | Summit |
| S1-2 | 30°09′04″ | 105°45′32″ | 320 | 15 | 1121.5 | 18.2 | Grassland | Shoulder |
| S1-3 | 30°09′03″ | 105°45′18″ | 307 | 10 | 1121.5 | 18.2 | Dry land | Backslope |
| S1-4 | 30°09′06″ | 105°45′19″ | 265 | 0 | 1121.5 | 18.2 | Paddy field to dry land | Footslope |
| S1-5 | 30°09′04″ | 105°45′16″ | 242 | 0 | 1121.5 | 18.2 | Paddy field to dry land | Toeslope |
| S2-1 | 29°57′56″ | 105°47′27″ | 330 | 2 | 1126.9 | 18.3 | Grassland | Summit |
| S2-2 | 29°57′56″ | 105°47′28″ | 328 | 2 | 1126.9 | 18.3 | Grassland | Shoulder |
| S2-3 | 29°57′56″ | 105°47′28″ | 314 | 10 | 1126.9 | 18.3 | Dry land | Backslope |
| S2-4 | 29°57′58″ | 105°47′31″ | 297 | 10 | 1126.9 | 18.3 | Dry land | Footslope |
| S2-5 | 29°57′56″ | 105°47′30″ | 265 | 0 | 1126.9 | 18.3 | Paddy field to dry land | Toeslope |
| S3-1 | 29°53′46″ | 105°46′59″ | 319 | 5 | 1130.3 | 18.5 | Paddy field | Summit |



| | | | | | | | | |
|---|---|---|---|---|---|---|---|---|
| | | | | | | | to dry land | |
| S3-2 | 29°53′46″ | 105°46′59″ | 301 | 10 | 1130.3 | 18.5 | Dry land | Shoulder |
| S3-3 | 29°53′48″ | 105°47′00″ | 298 | 6 | 1130.3 | 18.5 | Dry land | Backslope |
| S3-4 | 29°53′53″ | 105°47′00″ | 267 | 0 | 1130.3 | 18.5 | Dry land | Footslope |
| S3-5 | 29°53′55″ | 105°47′03″ | 240 | 0 | 1130.3 | 18.5 | Paddy field to dry land | Toeslope |
| S4-1 | 29°46′09″ | 105°40′25″ | 408 | 2 | 1126.7 | 18.0 | Grassland | Summit |
| S4-2 | 29°46′10″ | 105°40′24″ | 403 | 15 | 1126.7 | 18.0 | Dry land | Shoulder |
| S4-3 | 29°46′10″ | 105°40′22″ | 392 | 20 | 1126.7 | 18.0 | Dry land | Backslope |
| S4-4 | 29°46′10″ | 105°40′22″ | 396 | 0 | 1126.7 | 18.0 | Dry land | Footslope |
| S4-5 | 29°46′14″ | 105°40′20″ | 385 | 0 | 1126.7 | 17.9 | Dry land | Toeslope |
| S5-1 | 29°44′13″ | 105°40′19″ | 421 | 6 | 1127.6 | 18.0 | Grassland | Summit |
| S5-2 | 29°44′16″ | 105°40′19″ | 406 | 15 | 1127.6 | 18.0 | Dry land | Shoulder |
| S5-3 | 29°44′17″ | 105°40′18″ | 394 | 10 | 1127.6 | 18.0 | Dry land | Backslope |
| S5-4 | 29°44′17″ | 105°40′19″ | 396 | 0 | 1127.6 | 18.0 | Paddy field to dry land | Footslope |
| S5-5 | 29°44′18″ | 105°40′17″ | 387 | 0 | 1127.6 | 18.0 | Paddy field to dry land | Toeslope |

Notes: The average annual rainfall of the sampling points was calculated by the annual precipitation data of 1 km resolution in China (2001–2020) (National Earth System Science Data Center, National Science & Technology Infrastructure of China (http://www.geodata.cn)); The average annual temperature was calculated by the monthly mean air temperature raster data of China from 2001 to 2020 (1 km resolution) (He et al., 2021).

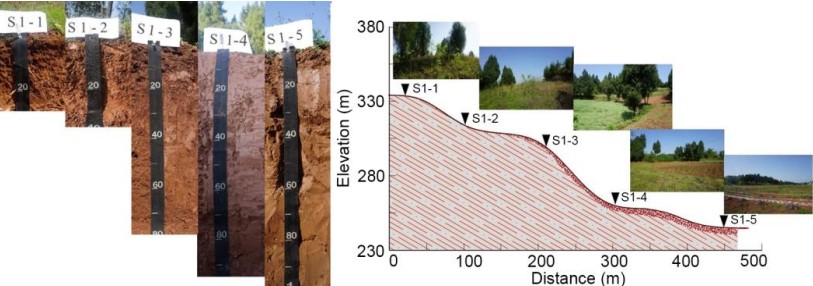

**Figure 2.** Soil profiles along a toposequence (S1-1, S1-2, S1-3, S1-4, and S1-5 represent the summit, shoulder, backslope, footslope, and toeslope positions of the hillslope, respectively. Take S1 as an example).

**2.3 Experimental method and data analysis**

The experimental methods used were the same as those used by Tang et al., (2019). In particular, the



soil bulk density was determined using the soil core (volume=100 cm³) method and the pipette method
was used for soil particle size analysis. A glass electrode was used to measure the soil pH in a 1:2.5
soil/water suspension ratio. Soil organic carbon (SOC) content was determined using the
dichromate-wet combustion method, and the C/N ratio was calculated as the ratio of SOC to total
nitrogen (TN) content (determined by the Kjeldahl method). Cation exchange capacity (CEC) was
determined using the Na saturation method. Each sample (0.5 g) was ground at 100 mesh (0.15 mm) in
an agate mortar and formulated into a tablet to measure the geochemical elements of the test soil using
X-ray fluorescence spectrometry (Sparks et al., 1996; Camobell et al., 2002). These geochemical
elements include macroelements, trace elements, rare-earth elements, and radioactive elements. The
content and composition characteristics of macroelements are widely used as indicators. Therefore, the
geochemical elements in this study refer to the 10 macroelements.
Statistical analyses were performed using SPSS 19.0, and included the analysis of paired sample
*T*-tests. The distribution map of the sampling sites was drafted using ArcGIS 10.2, and the other
diagrams were created using Excel 2016 and Origin 9.3.
**3 Results**
**3.1 Soil morphology**
As shown in Table 2, the profiles with the pattern of A-C horizons were mainly concentrated at the
summit and shoulder of the hillslope, the profile at the backslope and footslope was the A-B-C horizon,
and the toeslope was the A-B horizon within the excavation depth. From the summit to the toeslope,
the soil thickness increased significantly with the change in the soil profile configuration from 16.50 to
93.60 cm. In the study area, there were three types of soil colours, including 10R, 2.5YR, and 5YR.
The soil texture of horizon C was sandy loam or loam, whereas that of horizon B was mostly clay loam
with a few silty loams. The soil texture of horizon A was sandy or silty to clay loam, from the summit
to the toeslope (Table 3). The main soil structure of each horizon was blocky or/and granular, and the
organic matter accumulation in the surface soil, combined with mechanical ploughing, loosened the
soil. A granular structure appeared in the soil, and the structure improved. The cohesiveness of the soil
in horizon A was non-sticky, slightly sticky, and sticky from the summit to the toeslope. The





cohesiveness of horizon B was mostly sticky, whereas that of horizon C was mostly non-sticky. Owing
to human cultivation or mechanical tillage, there was a small amount of intrusive body in the soil,
mainly brick, tile debris, and a small amount of coal cinder. The parent rock of mudstone soil is
sedimentary rock, most of which is lacustrine facies sedimentary rock deposited in the Jurassic and
Cretaceous periods, therefore, a small number of shells were present in the soil.
**Table 2.** Morphological attributes of the soil profiles.

| Profile No. | Horizon | Depth (cm) | Soil colour | | Soil structure | Plasticity | Animal activity | Intrusions |
|---|---|---|---|---|---|---|---|---|
| | | | Dry state | Wet state | | | | |
| S1-1 | Ah | 0–12 | 2.5YR 5/6 | 2.5YR 4/6 | BS | Not plastic | Earthworm | - |
| | C | >12 | 2.5YR 5/6 | 2.5YR 4/6 | BS | Not plastic | - | - |
| S1-2 | Ap | 0–25 | 2.5YR 5/4 | 2.5YR 4/4 | GS | Not plastic | - | - |
| | C | >25 | 2.5YR 5/6 | 2.5YR 4/6 | BS | Not plastic | - | - |
| S1-3 | Ap | 0–18 | 2.5YR 6/4 | 2.5YR 4/4 | GS, BS | Not plastic | Ant nest | - |
| | Bw1 | 18–31/47 | 2.5YR 5/4 | 2.5YR 4/4 | BS | Slightly plastic | - | Bricks and rubbles |
| | Bw2 | 31/47–72 | 2.5YR 5/4 | 2.5YR 4/4 | BS | Slightly plastic | - | - |
| | C | >72 | 2.5YR 5/6 | 2.5YR 4/6 | BS | Not plastic | - | - |
| S1-4 | Ap | 0–12 | 2.5YR 5/3 | 2.5YR 4/3 | BS | Slightly plastic | - | Cinders |
| | Bw1 | 12–45 | 2.5YR 5/3 | 2.5YR 4/3 | BS | Medium plastic | - | Shells |
| | Bw2 | 45–70 | 2.5YR 5/3 | 2.5YR 4/3 | BS | Medium plastic | - | - |
| | Bw3 | 70–90 | 2.5YR 5/3 | 2.5YR 4/3 | BS | Medium plastic | - | - |
| | C | >90 | 2.5YR 5/6 | 2.5YR 4/6 | BS | Not plastic | - | - |
| S1-5 | Ap | 0–14 | 2.5YR 6/8 | 2.5YR 6/8 | BS | Slightly plastic | - | Bricks and rubbles |
| | Bw1 | 14–30 | 2.5YR 6/8 | 2.5YR 5/8 | BS | Medium plastic | - | Shells |
| | Bw2 | 30–58 | 2.5YR 6/8 | 2.5YR 5/8 | BS | Medium plastic | - | - |
| | Bw3 | 65–100 | 2.5YR 6/8 | 2.5YR 5/8 | BS | Plastic | - | - |
| S2-1 | Ah | 0–10 | 2.5YR 5/6 | 2.5YR 4/6 | BS | Not plastic | - | - |
| | C | >10 | 2.5YR 5/6 | 2.5YR 4/6 | BS | Not plastic | - | - |
| S2-2 | Ap | 0–15 | 2.5YR 5/4 | 2.5YR 5/6 | BS, GS | Not plastic | Earthworm, Centipede | - |
| | Bw | 15–40 | 2.5YR 5/4 | 2.5YR 4/4 | BS | Not plastic | - | - |



| Profile No. | Horizon | Depth (cm) | Soil colour Dry state | Soil colour Wet state | Soil structure | Plasticity | Animal activity | Intrusions |
|---|---|---|---|---|---|---|---|---|
| | C | >40 | 2.5YR 5/6 | 2.5YR 4/6 | BS | Not plastic | - | - |
| S2-3 | Ap | 0–24 | 2.5YR 6/6 | 2.5YR 6/6 | GS | Not plastic | Ant | - |
| | Bw1 | 24–35 | 2.5YR 6/6 | 2.5YR 5/6 | BS, GS | Not plastic | Beetle | - |
| | Bw2 | 35–51 | 2.5YR 6/6 | 2.5YR 5/6 | BS | Slightly plastic | - | - |
| | C | >51 | 2.5YR 5/6 | 2.5YR 4/6 | BS | Not plastic | - | - |
| S2-4 | Ap | 0–15 | 2.5YR 5/3 | 2.5YR 4/3 | BS | Not plastic | - | Cinders |
| | Bw1 | 15–35 | 2.5YR 5/3 | 2.5YR 4/3 | BS | Slightly plastic | - | Shells |
| | Bw2 | 35–62 | 2.5YR 5/3 | 2.5YR 4/3 | BS | Medium plastic | - | - |
| | Bw3 | 62–95 | 2.5YR 5/3 | 2.5YR 4/3 | BS | Medium plastic | - | - |
| | C | >95 | 2.5YR 5/6 | 2.5YR 4/6 | BS | Not plastic | - | - |
| S2-5 | Ap | 0–25 | 5YR 5/6 | 5YR 4/6 | BS, GS | Slightly plastic | - | - |
| | Bw1 | 25–42 | 5YR 5/4 | 5YR 4/4 | BS | Slightly plastic | - | Shells |
| | Bw2 | 42–70/74 | 5YR 5/4 | 5YR 4/4 | BS | Medium plastic | - | - |
| | Bw3 | 70/74–100 | 5YR 5/4 | 5YR 5/4 | BS | Medium plastic | - | - |
| S3-1 | Ah | 0–20/25 | 2.5YR 5/4 | 2.5YR 4/4 | BS | Not plastic | - | - |
| | C | >20/25 | 2.5YR 5/6 | 2.5YR 4/6 | BS | Not plastic | - | - |
| S3-2 | Ap | 0–20 | 2.5YR 5/4 | 2.5YR 6/6 | GS | Not plastic | Earthworm | - |
| | Bw1 | 20–33 | 2.5YR 5/4 | 2.5YR 6/6 | GS | Not plastic | - | - |
| | Bw2 | 33–60 | 2.5YR 5/4 | 2.5YR 6/6 | BS | Slightly plastic | - | - |
| | C | >60 | 2.5YR 5/6 | 2.5YR 4/6 | GS | Not plastic | - | - |
| S3-3 | Ap | 0–22 | 2.5YR 6/4 | 2.5YR 4/4 | GS | Not plastic | - | - |
| | Bw1 | 22–37 | 2.5YR 5/4 | 2.5YR 4/4 | BS, GS | Not plastic | - | Bricks |
| | Bw2 | 37–75 | 2.5YR 5/4 | 2.5YR 4/4 | BS | Slightly plastic | - | - |
| | C | >75 | 2.5YR 5/6 | 2.5YR 4/6 | BS | Not plastic | - | - |
| S3-4 | Ap | 0–15 | 2.5YR 5/3 | 2.5YR 4/3 | BS | Slightly plastic | Ant | - |
| | Bw1 | 15–38 | 2.5YR 5/3 | 2.5YR 4/3 | BS | Slightly plastic | - | - |
| | Bw2 | 38–75 | 2.5YR 5/3 | 2.5YR 4/3 | BS | medium | - | - |
| | Bw3 | 75–100 | 2.5YR 5/3 | 2.5YR 4/3 | BS | Medium | - | - |



| Profile No. | Horizon | Depth (cm) | Soil colour | | Soil structure | Plasticity | Animal activity | Intrusions |
|---|---|---|---|---|---|---|---|---|
| | | | Dry state | Wet state | | | | |
| S3-5 | Ap | 0–18 | 5YR 5/6 | 5YR 4/6 | BS | plastic / Slightly plastic | Earthworm | - |
| | Bw1 | 18–33 | 5YR 5/4 | 5YR 4/4 | BS | Medium plastic | - | - |
| | Bw2 | 33–56 | 5YR 5/4 | 5YR 4/4 | BS | Plastic | - | - |
| | Bw3 | 56–84 | 5YR 5/4 | 5YR 5/4 | BS | Plastic | - | - |
| | Bw4 | 84–100 | 5YR 4/4 | 5YR 3/4 | BS | Plastic | - | - |
| S4-1 | Ah | 0–20 | 10R 5/4 | 10R 4/4 | BS | Not plastic | - | - |
| | C | >20 | 10R 4/4 | 10R 3/4 | BS | Not plastic | - | - |
| S4-2 | Ap | 0–25 | 10R 5/4 | 10R 4/4 | GS | Not plastic | Earthworm | - |
| | Bw | 25–40 | 10R 4/4 | 10R 3/4 | GS | Slightly plastic | - | - |
| | C | >40 | 10R 4/4 | 10R 3/4 | BS | Not plastic | - | - |
| S4-3 | Ap | 0–20 | 2.5YR 5/6 | 2.5YR 4/6 | BS | Not plastic | - | - |
| | Bw | 20–40/55 | 2.5YR 5/6 | 2.5YR 4/6 | BS | Medium plastic | - | - |
| | C | >40/55 | 2.5YR 4/6 | 2.5YR 3/6 | BS | Not plastic | - | - |
| S4-4 | Ap | 0–20 | 2.5YR 5/3 | 2.5YR 4/3 | GS, BS | Not plastic | Ant | - |
| | AB | 20–35 | 2.5YR 4/3 | 2.5YR 4/3 | BS | Not plastic | - | - |
| | Bw1 | 35–51 | 2.5YR 4/3 | 2.5YR 4/3 | BS | Medium plastic | - | Cinders |
| | Bw2 | 51–83 | 2.5YR 4/3 | 2.5YR 4/3 | BS | Medium plastic | - | - |
| | C | >83 | 2.5YR 4/4 | 2.5YR 3/4 | BS | Not plastic | - | - |
| S4-5 | Ap | 0–20 | 2.5YR 5/3 | 2.5YR 4/3 | BS, GS | Not plastic | Earthworm | - |
| | Bw1 | 20–45 | 2.5YR 5/3 | 2.5YR 4/3 | BS | Medium plastic | - | Shells |
| | Bw2 | 45–68/75 | 2.5YR 5/3 | 2.5YR 4/3 | BS | Medium plastic | - | - |
| | Bw3 | 68/75–100 | 2.5YR 4/3 | 2.5YR 4/3 | BS | Medium plastic | - | - |
| S5-1 | Ah | 0–18 | 10R 5/6 | 10R 4/6 | GS | Not plastic | - | - |
| | C | >18 | 10R 4/6 | 10R 3/6 | BS | Not plastic | - | - |
| S5-2 | Ap | 0–10/15 | 10R 5/6 | 10R 4/6 | GS | Not plastic | Earthworm, Centipede | - |
| | AC | 10/15–25 | 10R 5/6 | 10R 4/6 | BS | Not plastic | - | - |
| | C | >25 | 10R 4/6 | 10R 3/6 | BS | Not plastic | - | - |
| S5-3 | Ap | 0–22 | 2.5YR 5/6 | 2.5YR 4/6 | BS | Not plastic | - | - |
| | AB | 22–45 | 2.5YR 5/6 | 2.5YR 4/6 | BS | Slightly | - | - |





| Profile No. | Horizon | Depth (cm) | Soil colour | | Soil structure | Plasticity | Animal activity | Intrusions |
|---|---|---|---|---|---|---|---|---|
| | | | Dry state | Wet state | | | | |
| | Bw | 45–65 | 2.5YR 5/6 | 2.5YR 4/6 | BS | plastic Medium plastic | - | Cinders |
| | C | >65 | 2.5YR 4/6 | 2.5YR 3/6 | BS | Not plastic | - | - |
| S5-4 | Ap | 0–10 | 2.5YR 5/3 | 2.5YR 4/3 | BS | Slightly plastic | Ant nest | Cinders |
| | Bw1 | 10–30 | 5YR 6/4 | 5YR 4/6 | BS | Medium plastic | - | Shells |
| | Bw2 | 30–55 | 2.5YR 4/3 | 2.5YR 4/3 | BS | Medium plastic | - | - |
| | Bw3 | 55–68 | 2.5YR 4/3 | 2.5YR 4/3 | BS | Plastic | - | - |
| | Bw4 | 68–100 | 2.5YR 4/4 | 2.5YR 3/4 | BS | Plastic | - | - |
| S5-5 | Ap | 0–30 | 5YR 5/6 | 5YR 4/6 | BS | Not plastic | Earthworm | - |
| | Bw1 | 30–55 | 5YR 5/4 | 5YR 4/4 | BS | Medium plastic | - | Shells |
| | Bw2 | 55–77 | 5YR 5/4 | 5YR 4/4 | BS | Medium plastic | - | - |
| | Bw3 | 77–100 | 5YR 5/4 | 5YR 5/4 | BS | Plastic | - | - |

Notes: BS, blocky structure; GS, granular structure.



As shown in Fig. 3, there were significant differences in the redness rating (RR) index between
different slope positions and horizons. According to the analysis of the RR index, slope position had a
significant effect on soil development. The RR index was relatively low in the whole region. The
maximum value of the RR index appeared in the parent material (C horizon), which indicated that
aluminium enrichment was weak in the process of soil development in this region, and the soil
development was relatively young. The RR index decreased from summit to toeslope, and that of the
footslope and toeslope were similar, with average values of 7.10 and 7.97, respectively. From the
profile horizon, the average RR index of the soil profile at different slope positions was in the
following order: horizon C > A > B (Fig. 3). The soil profile in horizon A was mainly affected by
farming, and that in horizon B was affected by topography. The soil redness rating varied greatly
among soil horizons, and the degree of soil development also differed.

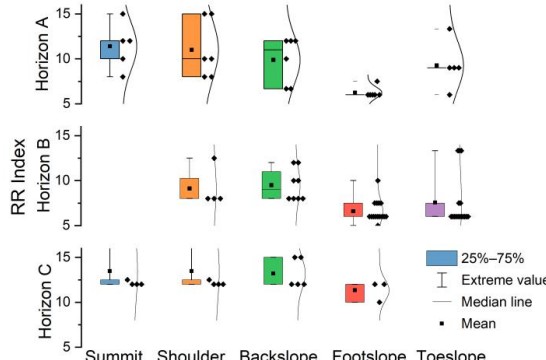


**Figure 3.** RR index of soil profiles in different slope positions.
**3.2 Soil physical and chemical properties**
The bulk density and porosity of the soil profiles at different slope positions were shown in Table 3.
The bulk density in horizons A and B of the soil profiles increased gradually from the summit to the
toeslope. The bulk density of the soil profiles at different slope positions showed that horizon A was
smaller than horizon B. The reason for this is that, during the process of soil development, the
aggregates of horizon A are in good condition with the accumulation of organic matter and an increase
in clay content. However, the soil in horizon B is compacted owing to disturbance during human
cultivation, and the bulk density then increases. The porosity in horizons A and B of the soil profiles



decreased gradually from the summit to the toeslope. The porosity of horizon A was higher than that of
horizon B. As shown in Table 3, the share of sand and silt fractions in horizons A and B decreased
gradually from the summit to the toeslope. The clay fraction content in horizons A and B increased
gradually from the summit to the toeslope.
**Table 3.** The bulk density, porosity, and particle size distribution of soil profiles at different slope
positions.

| Slope position | Horizon | Bulk density (g cm⁻³) | Porosity (%) | Particle size distribution (%) | | |
|---|---|---|---|---|---|---|
| | | | | Clay | Silt | Sand |
| Summit | A | 1.35±0.09 | 46.36±3.44 | 14.26±2.23 | 47.14±8.51 | 38.60±9.37 |
| | C | / | / | 10.99±3.73 | 40.21±1.88 | 48.80±2.95 |
| Shoulder | A | 1.37±0.11 | 45.5±4.82 | 17.25±2.72 | 44.65±3.33 | 38.09±4.95 |
| | B | 1.38±0.03 | 46.97±1.47 | 19.59±3.60 | 40.52±3.95 | 39.90±2.00 |
| | C | / | / | 10.39±1.88 | 39.26±1.53 | 50.35±3.14 |
| Backslope | A | 1.36±0.12 | 45.41±5.56 | 21.81±2.99 | 41.13±3.31 | 37.07±5.52 |
| | B | 1.48±0.14 | 42.41±5.20 | 22.14±4.08 | 39.90±5.30 | 37.96±5.89 |
| | C | / | / | 10.54±1.63 | 38.96±0.94 | 50.49±1.93 |
| Footslope | A | 1.43±0.04 | 43.32±1.80 | 30.82±4.19 | 39.33±1.96 | 29.85±4.01 |
| | B | 1.52±0.09 | 41.33±3.72 | 29.13±3.64 | 39.87±3.54 | 31.00±5.22 |
| | C | / | / | 10.13±1.52 | 39.60±0.98 | 50.27±2.11 |
| Toeslope | A | 1.53±0.16 | 39.64±6.09 | 36.47±3.47 | 38.79±1.96 | 24.74±4.20 |
| | B | 1.60±0.09 | 37.72±3.80 | 36.54±1.28 | 38.91±3.15 | 24.55±3.87 |

As shown in Fig. 4, the pH of horizon A and B was significantly higher at the summit, shoulder, and
backslope than that at the footslope and toeslope; however, there was no significant difference in the
pH of horizon C at different slope positions. The SOC content in horizon C was low, and there was no
significant difference among the different slope positions. The SOC content in horizons A and B
gradually increased with decreasing slope elevation and the difference in SOC above and below the
backslope was significant. The accumulation of SOC in horizon A at the footslope and toeslope was
mainly due to artificial cultivation and fertilisation, while the accumulation of SOC in horizon B was
mainly caused by detachment, transportation, accumulation, and burial of deep soil in the higher
topography. The hilly terrain was conducive to the accumulation of SOC content at the footslope and
toeslope. Part of the P in the soil comes from the parent rock, whereas the other part comes from the
application of chemical fertilisers and plant decomposition. As a highly mobile element, P easily
migrates in the lower horizon of the soil surface, whereas it is enriched in the surface horizon of soil
(Fig. 4). Total phosphorus (TP) in the soil at the summit of the slope was significantly lower than that



in the footslope and toeslope. The TP in the soil above the backslope was in the order of horizon B > C >
A, whereas that in the footslope and toeslope was A > B > C. The TN content from the summit to the
toeslope increased with decreasing slope elevation, and there was a significant difference above and
below the backslope, indicating that there was TN accumulation at the foot and toeslope. In general, the
TN above the backslope was in the order of horizon A > B > C of the soil profile, whereas that of the
footslope and toeslope was horizon B > A > C (Fig. 4). The total potassium (TK) content from the
summit to the toeslope gradually increased with increasing slope elevation. There was a significant
difference between the summit, shoulder of the slope and footslope, and toeslope, indicating that K in
the soil accumulated at a lower topographic position. According to the different horizons, the TK in the
soil above the backslope was in the order of horizon C > A > B, whereas that at the footslope was
horizon A > C > B (Fig. 4). There were differences in the CEC content among the different soil
horizons. The average CEC contents in horizons A, B, and C were 27.37, 29.67, and 28.36 cmol(+) kg$^{-1}$,
respectively. These results indicate that cations migrated and leached in horizon A and accumulated
and were enriched in horizon B. The CEC above the backslope was in the order of horizon C > B > A,
whereas that below the backslope was B > A > C.

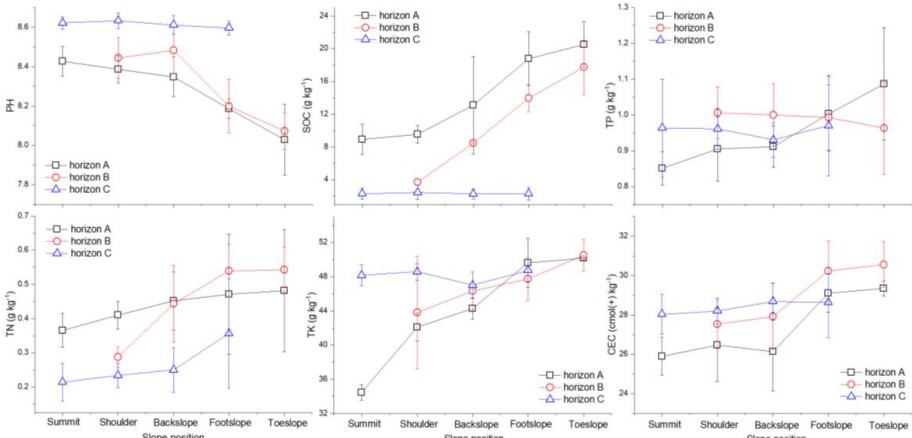


**Figure 4.** Chemical properties of soil profiles at different slope positions.
**3.3 Geochemical composition and chemical weathering indices**
In general, the footslope and toeslope of the hillslope showed accumulation of Al, Fe, Mg, and K,
and leaching loss of Ca, Na, and Si compared with the summit and shoulder of the hillslope. There was



a leaching loss of K in horizon A at the summit of the hillslope, while the variation in elements in each
horizon at the shoulder of the hillslope was not obvious (Fig. 5).
The leaching of Si and Na was obvious in horizon B of the slope profile, and the accumulation of Al,
Fe, and Mg was significant in horizon B of the slope profile relative to horizon C. The leaching of Na
and Ca was significant in horizons A and B, whereas the leaching of Ca and Na was obvious in horizon
B relative to horizon A in the hillslope. During the development of silicate soils, Na and Ca were
leached first, followed by K, Si was leached in the late stage, and Al and Fe were relatively enriched.
The soil in this study was in the stage of leaching of Ca and Na, the leaching of soil geochemical
elements was not strong, and the degree of soil development was weak. The effect of microtopography
on soil chemical weathering was greater than that on the profile.

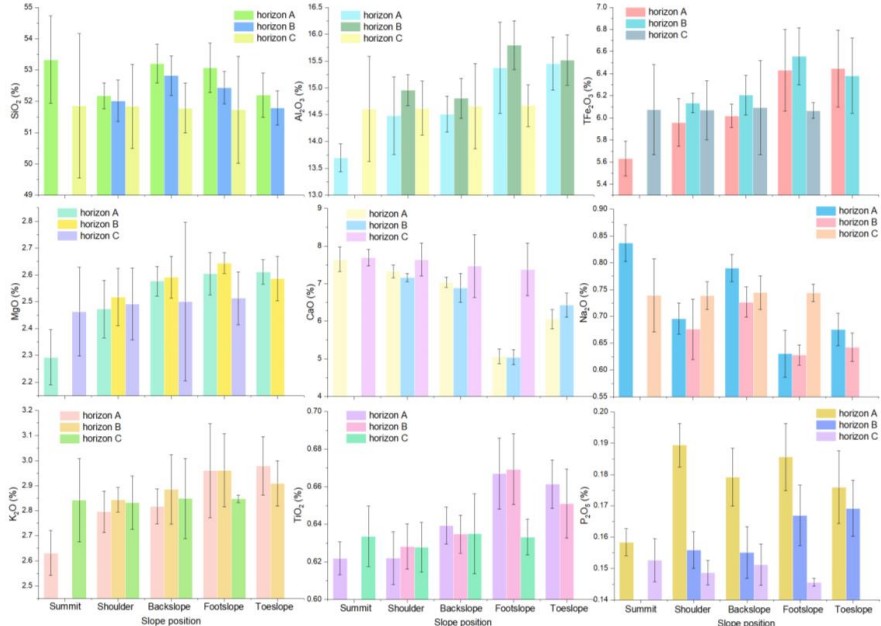


**Figure 5.** Geochemical element contents of soil profiles at different positions.
The chemical index of alteration (CIA) values were 71.78, 73.17, 72.51, 74.55, and 74.55 from
summit to toeslope. Moreover, there were significant differences in CIA values between horizons A
and B at the summit, shoulder, backslope, footslope, and toeslope. The CIA values of horizons A, B,
and C were 72.92, 74.37, and 72.61, respectively. The variation trend of chemical index of weathering
(CIW) was basically the same as that of CIA, while the indication effects of CIA and CIW to reflect the
degree of soil weathering were the same. Correlation analysis of CIA, CIW, and Na/K showed that



there was a significant negative correlation between Na/K and CIA and CIW (Fig. 6). The CIA, CIW,
and Na/K values indicated that the soil in the study area had moderate chemical weathering, and the
chemical weathering of the soil parent horizon remained at the same level from the summit to the
toeslope of the hillslope. Horizons A and B showed a trend of first increasing and then decreasing with
decreasing slope elevation, and the soil of horizon B at the footslope had the highest degree of
development, which indicated that the soil was strongly detached and transported under the action of
microtopography, and the deeply weathered soil in higher terrain was deposited into the soil of horizon
B at the footslope.

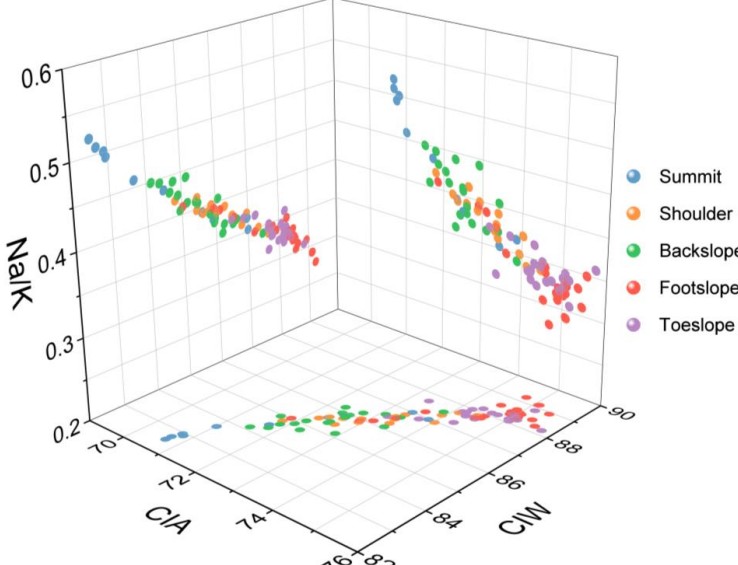


**Figure 6.** Chemical weathering parameters of soil profiles at different positions.

As shown in Fig. 7, the mobility of the elements varies at different microtopographical positions.

The migration directions of Ca and Na at the summit and backslope were completely opposite to that at
the footslope. The average migration coefficients of Ca at the summit, backslope, and footslope were
1.43%, -4.93%, and -33.43%, respectively, while the average migration coefficients of Na were
16.15%, 1.73%, and -18.43%, respectively, and the migration direction changed from enrichment to
leaching. Al, Fe, and Mg were first leached and then enriched from the summit to the footslope, which
may be due to the higher sand content in the soil and relatively large intergranular pores, resulting in
the loss of elements under the action of rainfall and underground runoff. However, where the terrain
was relatively low, such as the footslope, the water condition was sufficient, which provided conditions





for soil chemical weathering, and Ca and Na leached out, while Al and Fe were enriched.

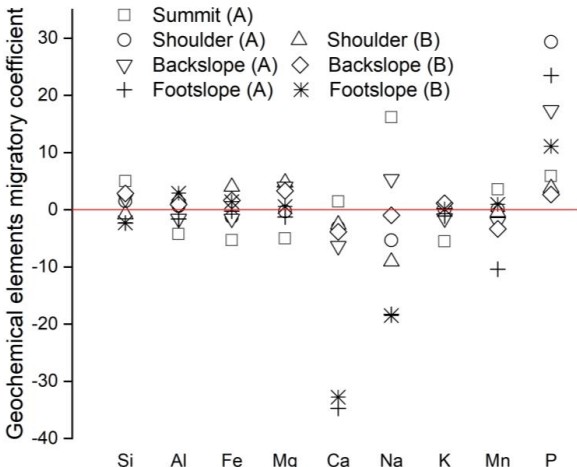


**Figure 7.** Geochemical elements migratory coefficient of soil profiles at different slope positions.
**3.4 Mineralogy characteristics**

The mineral composition of the soils was analysed using X-ray diffraction (Fig. 8). There was an

evident mineralogical similarity between the soils at different slope positions and horizons. The
mudstone soils in this study were dominated by illite, kaolinite, and an illite/smectite mixed horizon,
and the mineral composition of the soils was the same as that of their parent rock. These results
indicate that the clay minerals in the soils originated mostly from the parent material and that they were
only slightly influenced by soil-forming processes.

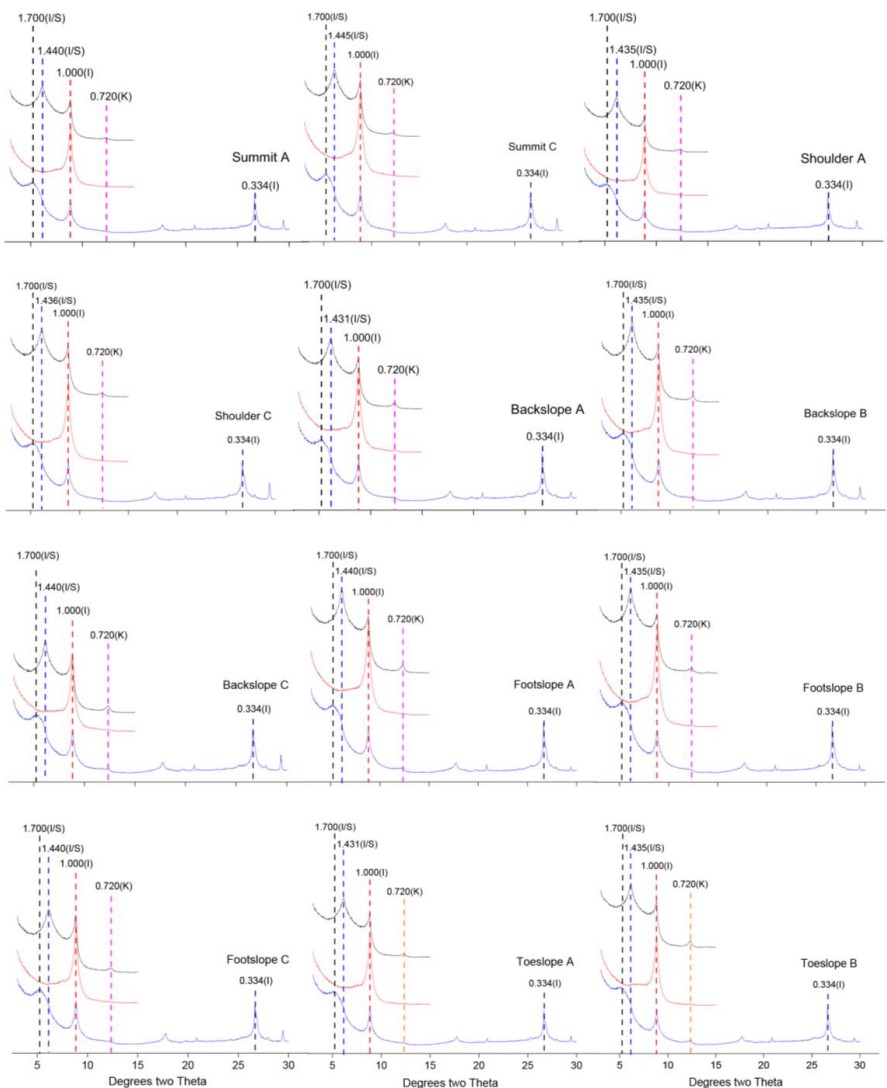

**Figure 8.** X-ray diffraction data for soils at different slope positions (A−horizon A, B−horizon B, and
C−horizon C). The sequence of treatments is represented by three different colours: the black (top), red
(middle position), and blue (bottom) curves represent the air-dried mount (AD) samples, samples
heated at 550 ℃ (K550), and glycol solvated (EG) samples, respectively. I: illite; K: kaolinite; I/S:
illite/smectite mixed-horizon.

**4 Discussion**





**4.1 Microtopography affects the chemical weathering of the soil at different slope positions**


As shown in Table 4, there was a significant negative correlation between sand content and SOC,
TN, TK, CEC, and bulk density and a significant positive correlation between sand content and pH and
porosity. There was a significant negative correlation between silt content and SOC, TN, TP, TK, and
CEC, and a significant positive correlation between silt content and pH. There was a negative
correlation between clay content and pH and porosity, and a significant positive correlation between
clay content and SOC, TN, TK, CEC, and bulk density. There is a significant correlation between the
particle composition and amorphous Fe and Al content (Obi, 2015), particularly the clay content.
Amorphous Fe can provide a high specific surface area to influence ion adsorption in soil, whereas Al
participates in hydrolysis to that influence soil pH (Harter, 2007). The lower alkali metals in the soil
increased the $H^+$ on the mineral surface, resulting in the destruction of the mineral structure and the
release of other mineral ions, such as $Si^{4+}$, $Fe^{3+}$, and $Al^{3+}$. The released $Fe^{3+}$ and $Al^{3+}$ formed hydrated
oxides under appropriate conditions and remained in the soil solution. Finally, the amorphous hydrated
oxides of aluminium and iron, together with kaolinite, become the main solids in the soil. That is,
feldspar is rearranged by Si and Al under weathering to form clay minerals such as kaolinite, which
explains the relationship between particle composition and hydrated oxides. Therefore, the particle
composition, particularly the clay content, is an important variable that affects the decisive factors of
soil characteristics.
**Table 4**. Pearson's correlation coefficients of soil physical and chemical properties.

| | SP | pH | SOC | TN | TP | TK | CEC | BD | Porosity | Sand | Silt | Clay |
|---|---|---|---|---|---|---|---|---|---|---|---|---|
| SP | 1 | | | | | | | | | | | |
| pH | -0.328** | 1 | | | | | | | | | | |
| SOC | 0.170 | -0.605** | 1 | | | | | | | | | |
| TN | 0.172 | -0.530** | 0.688** | 1 | | | | | | | | |
| TP | 0.079 | -0.221 | 0.156 | 0.191 | 1 | | | | | | | |
| TK | 0.419** | -0.609** | 0.500** | 0.496** | 0.373** | 1 | | | | | | |
| CEC | 0.469** | -0.422** | 0.532** | 0.433** | 0.173 | 0.564** | 1 | | | | | |
| BD | 0.400** | -0.354** | 0.382** | 0.363** | 0.088 | 0.522** | 0.436** | 1 | | | | |
| Porosity | -0.323** | 0.357** | -0.422** | -0.370** | -0.054 | -0.478** | -0.354** | -0.977* | 1 | | | |
| Sand | -0.255* | 0.531** | -0.618** | -0.428** | -0.086 | -0.515** | -0.500** | -0.557** | 0.565** | 1 | | |
| Silt | -0.265* | 0.340** | -0.277* | -0.300* | -0.275* | -0.457** | -0.356** | -0.100 | 0.054 | -0.161 | 1 | |
| Clay | 0.383** | -0.680** | 0.725** | 0.562** | 0.232 | 0.730** | 0.661** | 0.573** | -0.555** | -0.839** | -0.402** | 1 |

Notes: SP, slope position; BD, bulk density; *, $p \le 0.05$; **, $p \le 0.01$.
The degree of soil chemical weathering varied with changes in the slope position and soil depth.



According to the variance $F$ value of the soil chemical weathering index under different factors (Table
5), the CIA, CIW, and Na/K of soils varied significantly between different horizons at different slope
positions and the interaction between slope position and horizon. Therefore, topographic changes
caused differences in chemical weathering of the soil profile, mainly because the soil at the top and
shoulder of the slope was mostly developed under natural conditions, and soil weathering mainly
depended on the change in natural hydrothermal status and the effect of soil organisms. The return of
nutrient elements such as Na, Ca, and Mg by litter may decelerate the chemical weathering process,
thus decelerating soil development in the middle and upper hills. At the footslope and other relatively
low topographic locations, soil moisture is retained, which provides moisture conditions for chemical
weathering. The depth of the soil profile at the footslope was more than 1 m, and a small or moderate
number of rust spots began to appear in the soil horizon, resulting in changes in the soil profile
morphology and chemical characteristics. Meanwhile, the soil on the footslope and toeslope was
affected by human cultivation. Irrigation changes the soil moisture status, tillage changes soil aeration
conditions (Wei et al., 2006), and fertilisation increases soil nutrient elements. These factors promote
soil mineral weathering and element leaching, and soil erosion intensifies (He et al., 2007; Ni and
Zhang, 2007; Zheng et al., 2007). Therefore, soil erosion caused by tillage promotes chemical
weathering in the soil. In areas with large terrain slopes, the degree of physical rock erosion is greater,
and chemical weathering is stronger (Gabet, 2007; Riebe et al., 2004).
**Table 5.** Analysis of variance $F$ value of chemical weathering indicators of soils at different slope
positions and horizons.

| Factor | CIA | | CIW | | Na/K | |
|---|---|---|---|---|---|---|
| | $F$ | Sig. | $F$ | Sig. | $F$ | Sig. |
| Slope position | 27.298 | 0.000 | 29.754 | 0.000 | 23.191 | 0.000 |
| Occurrence layer | 19.951 | 0.000 | 16.077 | 0.000 | 8.985 | 0.000 |
| S × O | 7.222 | 0.000 | 9.223 | 0.000 | 8.506 | 0.000 |

Notes: S × O, interaction of slope position and horizon.
**4.2 The redistribution of water and materials by microtopography resulted in the difference of**
**pedogenic characteristics at different slope positions**
Water flow runs through the entire process of soil occurrence and development. Water plays an
important role in soil formation as the main medium for transporting solids and ions in the soil
(Schaetzl and Anderson, 2005). Microtopography dominates surface hydrological processes and affects



water redistribution (Muscarella et al., 2020; Peñuela et al., 2015; Wang et al., 2022), resulting in
differences in the physicochemical properties of soil at different slope positions (Maranhão et al., 2020).
Therefore, even in a microdomain of tens of meters, the soil of the same genus also forms different
textures owing to the difference in properties. In this study, the physicochemical properties of the soil
varied with the change in the slope position, particularly between the soil above and below the
backslope, and there were also differences among the different horizons. From the summit to toeslope,
the sand content of the soil gradually decreased, and the clay content of the soil gradually increased.
The mudstone parent rock exposed at the summit of the hilly mountainous region caused by erosion is
rich clay minerals and has strong water absorption capacity, therefore, it is easily physically weathered
under the influence of moist heat expansion. Consequently, stony subsoil is frequently formed at or
near the summit of the slope (Zhong et al., 2020). Obi et al., (2014) showed that sand content is
affected by rainfall and infiltration. When rainfall exceeded infiltration, it caused redistribution of sand
in the slope and soil horizon. This phenomenon is often observed in thunderstorm weather, which
reduces the stability of soil on the slope surface (Eneje et al., 2007; Oguike and Mbagwu, 2009). The
results of this study also showed that the degree of chemical weathering of horizon A at the footslope
and toeslope was lower than that of horizon B, which was mainly due to the redistribution of soil
materials by microtopography (Baltensweiler et al., 2020). Soil and water loss is serious in hilly
mountainous regions, and the materials transported by upper erosion are deposited at the footslope and
toeslope. Long-term contact between water and sediment leads to further chemical action, resulting in
soil with high organic matter content and fine texture, which is buried by a new round of denudation
accumulation and self-weathered soil, becoming a B-horizon soil with a higher degree of development
than the topsoil.
**5 Conclusion**
Five soil profiles along the slopes were sampled at the summit, shoulder, backslope, footslope, and
toeslope positions on behalf of the microtopography. The morphological characteristics,
physiochemistry, and geochemical attributes of the profiles were analysed. The results indicate that the
morphological characteristics of the mudstone soil profile were mainly inherited and affected by the
parent material. There was a significant correlation between the soil and parent material developed at



different slope positions. Under the effect of topography, different parts of the hillslope exhibited
different profile morphologies. From the summit to the toeslope, the profile configuration of the
mudstone soil changed from A-C to A-B-C, and the thickness of the soil increased. The bulk density,
clay fraction, soil organic matter, TN, TP, TK, and CEC increased from the summit to the toeslope of
the hillslope, whereas the pH, porosity, sand, and silt fraction decreased. From the perspective of soil
element migration, the migration of geochemical elements differed at different topographic locations.
The migration direction of Ca and Na at the summit, backslope, and footslope changed from
enrichment to leaching, whereas that of Al, Fe, and Mg changed from leaching to enrichment. In
addition to the parent material, the development of mudstone soil is significantly related to
microtopographic action. At the summit and shoulder of the hillslope, weathered materials are
transported away by gravity and surface erosion, and new rocks are often exposed; therefore the
characteristics of soil development is relatively weak. Alternatively, in flat areas such as the footslope
and toeslope with sufficient water conditions, the long-term contact between water, soil, and sediment
leads to further chemical weathering, resulting in a higher degree of soil development.
Microtopography can affect physicochemical properties, chemical weathering, and redistribution of
water and materials, resulting in differences in pedogenic characteristics at different slope positions.
**Data availability**
All raw data can be obtained from the corresponding authors on request.
**Author contributions**
Conceptualization, Banglin Luo and Jiangwen Li; investigation and data curation, Banglin Luo,
Jiangwen Li and Jiahong Tang; methodology and formal analysis, Banglin Luo, Jiangwen Li and
Jiahong Tang; writing-original draft preparation, Banglin Luo; writing-review and editing, Jiangwen Li,
Chaofu Wei and Shouqin Zhong; project administration and supervision, Chaofu Wei and Shouqin
Zhong; funding acquisition, Chaofu Wei and Shouqin Zhong.
**Competing interests**
The authors declare that they have no conflict of interest.



**Acknowledgements**
This study was supported by the Fundamental Research Funds for the National Natural Science
Foundation of China (42077007), and the Startup Project of Doctor Scientific Research at Southwest
University (No. SWU120065). Acknowledgement for the data support from "National Earth System
Science Data Center, National Science & Technology Infrastructure of China.
(http://www.geodata.cn)".

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
