# Peer review of "How do Microtopography Act in the Pedogenic"

_EGUsphere, 2022_

## Referee Comment (RC2)

Review of egusphere-2022-1352

General comments: I would like to thank Luo et al. for their submission to SOIL. I enjoyed reading about toposequences in this region of China. The authors presented a suit of physical, chemical, and mineralogical data for 5 toposequences forming from similar parent material in the Sichuan basin. They general found changes in bulk elemental composition with slope position. The current state of the manuscript is not ready for publication and needs major revision. Currently, this manuscript presents a lot of data but lacks organization and direction in terms of the manuscript objectives, findings, and discussion.

First, why the use of "microtopography"? When I think of microtopography, my references scale is maybe a meter to 3 meters, I think of swale topography or tree throw. Here, this is hillslope scale topography, you could use "toposequence" or more classically "catena" to describe the scale upon which the work is relevant. I would also prefer the title be written, currently is grammatically incorrect. Just write as a statement describe the major finding of the work: "Topographic influences on pedogenesis in the mudstone derived soils of the Sichuan basin".

The introduction is general and unorganized. You could better hone this by starting more generally presenting how topography impacts soil formation and why on a small scale it overrides other soil forming factors. Then becoming more specific and reviewing other relevant examples of toposequences, their findings, and how there is a gap in the current literature that your work will ultimately fill. Finally, the manuscripts objectives should be clarified. Currently, I think the three that are listed are just different ways of saying the same thing: we are going to investigate how landscape position influences soil formation. You have a lot of data, you also discuss some of the human influences that are landscape dependent, which I think is a potential interesting way to investigate human activities as a soil forming factor. What other objectives/hypotheses could be developed and tested using the data that you already have?

The methods appear incomplete. In terms of sites, do you select sites that all had the same aspect? And if so, what were they? You cited a reference and stated that you followed similar methods. You should still provide the reader and relatively sufficient summary that they could reasonably understand what you did and replicate your study. You later presented mineralogy data but added no mineralogy prep methods, this should be added to the methods section. You also wrote certain sections that reference significance without statistics or describing how stats like ANOVA and person's correlations were calculated. These methods and statistical techniques should be added to the methods section.

The discussion is mainly just generalities about soil formation with little discussion of the results in context of the available literature. Once you reformulated and thought through your hypotheses, you can use them to guide your discussion. Did you confirm the hypothesis? If so, what does that mean for soil formation? If not, what else may be occurring that could explain your results. Providing context from other toposequences would be highly valuable. What have others found in similar toposequences? Do they align with your results? Do they not? Why or why not? This discussion would provided greater context for your findings.

Specific comments:

Line 14: delete "the" at beginning of sentence.

Line 19: Do you mean "deposits"? not 'deposition'?

Line 19-20: Could you provide more relevant geographic information? For example the approx. distance from the site to dominant city in the Sichuan basin?

Line 22-24: Revise sentence to: "From the summit to the toeslope, soil thickness increased significantly and profile configuration changed from A-C to A-B-C."

Line 24-26: I do not currently understand how the patterns in Ca, Na, Al, Fe, and Mg changed as written. Are you saying that you find that in the summit Ca and Na were enriched in the soil relative to the PM but was depleted in the Footslope? Same goes for Al, Fe, and Mg.

Line 29-31: You also have the addition of weathered components and I think from the previous sentence enrichment of Al, Fe, Mg. Longer residence time for sure, but also additions of these elements.

Line 82-85: As written, these aims are the same. You are equating slope positions with microtopography, correct?

Line 86: How does this work lead to regulation of soil forming processes? Are you trying to say that you could use this work as a basis for soil management?

Line 127-128: Did you perform any pretreatments for your PSA? Like removal OM?

Line 149: Should be 'silt loam' not 'silty loam'.

Table 2: You can't use the 'h' suffix with the A horizon. I'm not sure what you are saying with the "Ah" horizons, which is consistent across your summit positions.

Line 161: You need to cite Torrent et al. (1983) for the redness rating calculation.

Line 163: I do not agree that the RR you have calculated is "low", 10R and 2.5YR hues are really red, I think that you should think more carefully about the RR values in light of the soil colors you have described in Table 2.

Line 164: So the max RR is in the C horizon, so could you have a red parent material and the soil is just inheriting this color. If this is the case, is RR/color a good indicator of soil development?

Line 175: How did you assess porosity? Is this just assuming a certain mineral density? Or did you evaluate this in the lab? Please provided these methods in the methods section.

Line 177: Do mean the bulk density of the A horizons was less than the B horizons?

Line 178 – 181: Your reasoning about the differences in bulk density between A and B horizons is incorrect. The A horizons have a lower bulk density because they have higher OM and from your texture more clay content, which has a lower density than inorganic minerals. The B horizons likely have a higher bulk density due to compaction from the overlying A horizon mass, a loss of OM, and an increase with sand content. Also, how could it be that the A horizons are not also disturbed by the human activity but the underlying B horizons are? Are you trying to suggest this is a plow pan?

This should also be moved to the discussion section.

Line 188: When you say significantly higher, did you do a statistical test to test this? Please report your p values and test statistic here. This applies to the entire results section.

Line 232, Fig 5: Large easy to read text indicating which elements is being present. I also think this figure might be more informative as a table.

Line 233-238: How do you know that there were significant differences, please present some statistics to help us evaluate these differences.

Line 238-239: Again, please report statistics for these correlations.

Line 239-246: What figure are you referencing when discussing the trends in CIA, CIW, and Na/K changing between landscape positions within the A, B, and C horizons?

Ling 244: What do you mean by "strongly detached"? I think that you are just saying that the B horizon is comprised of eroded and weathered material from higher landscape positions.

Line 251-254: What are the migration coefficients and how are they calculated? Are these tau values? Or some other indicator of element enrichment/depletion?

Line 262-267: How did you prep your samples for mineralogy? Are these oriented clay slides? What XRD instrument and scan parameters did you use. What treatments did you use to differentiate minerals?

Line 276: Please provide these statistical techniques in the methods section.

Lines 276-281: This should be moved to the result section.

Lines 283-292: How does your data provide a measure of Fe/Al oxide content/concentrations? Did you perform dithionite or oxalate extractions to assess the presence of crystalline/amorphous Fe/Al oxides? Further, this is rather general review information and does not provide a specific context for your results? Can you dig deeper into your data to provide some context for topographic influences on the mineralogy and geochemistry that you presented?

Lines 296-298: This is redundant to what is reported in the results in lines 239-246. Please delete here and report statistics in the results section.

Line 298-301: I'm really not sure what the sentence is saying. How are the soils at the summit and shoulder positions developed under natural conditions whereas other slope positions are not? Are you trying to say these have experienced less human influences? Please explain what you mean by 'natural hydrothermal status'. Are these soils impacted by hydrothermal (i.e., heated groundwater) processes? If so, that substantial changes some of the interpretations about your data.

Line 302: Sodium is not a necessary nutrient for most plants. For this reason, we often use it as an indicator of chemical weathering and leaching because it is not biocycled. I also don't think that cycling

of Ca, Mg would slow chemical weathering, it could change the distribution of Ca and Mg within soil profile, but there will always be some leaching and loss from the soil.

Line 306: Instead of rust spots, do you mean redoximorphic features?

Line 308-312: I think it is a big leap to say that tillage causes greater chemical weathering. If you wanted to discuss this potential it would have been informative to include a representative catena that is largely unimpacted by humans.

Line 312-313: Unnecessary sentence. Please delete.

Table 5: Move to the results please.

Section 4.2: Could you compare your study to other toposequence studies? I'm sure there are relevant examples of other subtropical monsoonal toposequences? Maybe these are even in similar parent material or different aspects to compare?

---

## Author Comment (AC5)

***Q1:*** First, why the use of "microtopography"? When I think of microtopography, my references scale is maybe a meter to 3 meters, I think of swale topography or tree throw. Here, this is hillslope scale topography, you could use "toposequence" or more classically "catena" to describe the scale upon which the work is relevant. I would also prefer the title be written, currently is grammatically incorrect. Just write as a statement describe the major finding of the work: "Topographic influences on pedogenesis in the mudstone derived soils of the Sichuan basin".

***Response***: Thanks very much for your constructive comments. Regarding "microtopography", we initially referenced several literatures (Baltensweiler et al., 2020; Lv et al., 2023; Maranhão et al., 2020; Pal et al., 2003; Thompson et al., 2010; Wang et al., 2022) and defined it as the small-scale hilly area with little relief to differentiate it from the large terrain sequence. The aim of this study is to explain how mudstone is affected by microtopography during the process of soil formation in this hilly area with little relief; thus, the term "microtopography" is appropriate. After careful consideration, we agree that it is a small-scale terrain sequence, and we will adjust the terminology accordingly based on your suggestions.

The references as follow:

Baltensweiler, A., Heuvelink, G. B. M., Hanewinkel, M., and Walthert, L.: Microtopography shapes soil pH in flysch regions across Switzerland, Geoderma, 380, 114663, https://doi.org/10.1016/j.geoderma.2020.114663, 2020.

Lv, W., Liu, Y., Du, J., Tang, L., Zhang, B., Liu, Q., Cui, X., Xue, K., and Wang, Y.: Microtopography mediates the community assembly of soil prokaryotes on the local-site scale, CATENA, 222, 106815, https://doi.org/10.1016/j.catena.2022.106815, 2023.

Maranhão, D. D. C., Pereira, M. G., Collier, L. S., Anjos, L. H. C. dos, Azevedo, A. C., and Cavassani, R. de S.: Pedogenesis in a karst environment in the Cerrado biome, northern Brazil, Geoderma, 365, 114169, https://doi.org/10.1016/j.geoderma.2019.114169, 2020.

Pal, D. K., Srivastava, P., Durge, S. L., and Bhattacharyya, T.: Role of microtopography in the formation of sodic soils in the semi-arid part of the Indo-Gangetic Plains, India, CATENA, 51, 3–31, https://doi.org/10.1016/S0341-8162(02)00092-9, 2003.

Thompson, S. E., Katul, G. G., and Porporato, A.: Role of microtopography in rainfall-runoff partitioning: An analysis using idealized geometry, Water Resour. Res., 46, https://doi.org/10.1029/2009WR008835, 2010.

Wang, Y., Li, S., Lang, X., Huang, X., and Su, J.: Effects of microtopography on soil fungal community diversity, composition, and assembly in a subtropical monsoon

evergreen broadleaf forest of Southwest China, CATENA, 211, 106025, https://doi.org/10.1016/j.catena.2022.106025, 2022.

*Q2:* The introduction is general and unorganized. You could better hone this by starting more generally presenting how topography impacts soil formation and why on a small scale it overrides other soil forming factors. Then becoming more specific and reviewing other relevant examples of toposequences, their findings, and how there is a gap in the current literature that your work will ultimately fill. Finally, the manuscripts objectives should be clarified. Currently, I think the three that are listed are just different ways of saying the same thing: we are going to investigate how landscape position influences soil formation. You have a lot of data, you also discuss some of the human influences that are landscape dependent, which I think is a potential interesting way to investigate human activities as a soil forming factor. What other objectives/hypotheses could be developed and tested using the data that you already have?

*Response*: Thanks very much for your suggestion. We will rewrite the introduction according to your comments.

*Q3:* The methods appear incomplete. In terms of sites, do you select sites that all had the same aspect? And if so, what were they? You cited a reference and stated that you followed similar methods. You should still provide the reader and relatively sufficient summary that they could reasonably understand what you did and replicate your study. You later presented mineralogy data but added no mineralogy prep methods, this should be added to the methods section. You also wrote certain sections that reference significance without statistics or describing how stats like ANOVA and person's correlations were calculated. These methods and statistical techniques should be added to the methods section.

*Response*: Thanks very much for your suggestion. We have updated the "Materials and Methods" section to include additional information on soil mineralogy prep methods, soil porosity measurement methods, and methods for determining statistical significance, etc.

*Q4:* The discussion is mainly just generalities about soil formation with little discussion of the results in context of the available literature. Once you reformulated

and thought through your hypotheses, you can use them to guide your discussion. Did you confirm the hypothesis? If so, what does that mean for soil formation? If not, what else may be occurring that could explain your results. Providing context from other toposequences would be highly valuable. What have others found in similar toposequences? Do they align with your results? Do they not? Why or why not? This discussion would provided greater context for your findings.

*Response*: Thanks very much for your constructive comments. We will make modifications based on your comments and discuss our findings in depth by comparing similar toposequences from other studies with existing literatures.

Specific comments:

*Q5:* Line 14: delete "the" at beginning of sentence.

*Response*: Thanks for your suggestion. We deleted "the" at beginning of sentence.

*Q6:* Line 19: Do you mean "deposits"? not 'deposition'?

*Response*: Yes. We have changed "deposition" into "deposits".

*Q7:* Line 19-20: Could you provide more relevant geographic information? For example the approx. distance from the site to dominant city in the Sichuan basin?

*Response*: Thanks for your suggestion. We have supplemented the approximate distance from the sampling sites to the main city zone of Chongqing.

*Q8:* Line 22-24: Revise sentence to: "From the summit to the toeslope, soil thickness increased significantly and profile configuration changed from A-C to A-B-C."

*Response*: Thanks for your suggestion. We have revised the sentence.

*Q9:* Line 24-26: I do not currently understand how the patterns in Ca, Na, Al, Fe, and Mg changed as written. Are you saying that you find that in the summit Ca and Na were enriched in the soil relative to the PM but was depleted in the Footslope? Same goes for Al, Fe, and Mg.

*Response*: Yes. Thank you for your suggestion. What we want to express is that the content of Ca and Na is relatively higher at the summit compared with the footslope, due to continuous leaching from the summit to the footslope. Conversely, the content of Al, Fe and Mg exhibited an inverse trend. We have made revisions to this sentence.

*Q10:* Line 29-31: You also have the addition of weathered components and I think

from the previous sentence enrichment of Al, Fe, Mg. Longer residence time for sure, but also additions of these elements.

*Response*: Yes. Thank you for your suggestion. We have revised the sentence.

*Q11:* Line 82-85: As written, these aims are the same. You are equating slope positions with microtopography, correct?

*Response*: Thank you for your suggestion. We will rewrite the research objectives according to your comments.

*Q12:* Line 86: How does this work lead to regulation of soil forming processes? Are you trying to say that you could use this work as a basis for soil management?

*Response*: Yes. Thank you for your suggestion. We have revised the sentence.

*Q13:* Line 127-128: Did you perform any pretreatments for your PSA? Like removal OM?

*Response*: Thanks for your question. The organic matter (OM) was removed from the soil before conducting the particle size analysis. We have added additional steps to the main process of conducting the PSA.

*Q14:* Line 149: Should be 'silt loam' not 'silty loam'.

*Response*: Thanks for your suggestion. We have revised the expression in this paper.

*Q15:* Table 2: You can't use the 'h' suffix with the A horizon. I'm not sure what you are saying with the "Ah" horizons, which is consistent across your summit positions.

*Response*: Thanks for your suggestion. The "Ah" horizon is primarily used to indicate the illuvial accumulation of organic matter on the slope of grassland at the summit.

*Q16:* Line 161: You need to cite Torrent et al. (1983) for the redness rating calculation.

*Response*: Thanks very much for your suggestion. Based on your suggestion and after careful consideration, we have determined that a high RR value of weathered parent material is not a suitable indicator for soil development. Therefore, we have removed it from our study.

*Q17:* Line 163: I do not agree that the RR you have calculated is "low", 10R and 2.5YR hues are really red, I think that you should think more carefully about the RR values in light of the soil colors you have described in Table 2.

*Response*: Thanks for your suggestion. We have removed the RR values from our study.

*Q18:* Line 164: So the max RR is in the C horizon, so could you have a red parent material and the soil is just inheriting this color. If this is the case, is RR/color a good indicator of soil development?

*Response*: Thanks for your suggestion. We have removed the RR values from our study.

*Q19:* Line 175: How did you assess porosity? Is this just assuming a certain mineral density? Or did you evaluate this in the lab? Please provided these methods in the methods section.

*Response*: Thanks very much for your suggestion. We have added further methods for measuring soil porosity to the materials and methods section.

*Q20:* Line 177: Do mean the bulk density of the A horizons was less than the B horizons?

*Response*: Yes. We have revised the expression.

*Q21:* Line 178 – 181: Your reasoning about the differences in bulk density between A and B horizons is incorrect. The A horizons have a lower bulk density because they have higher OM and from your texture more clay content, which has a lower density than inorganic minerals. The B horizons likely have a higher bulk density due to compaction from the overlying A horizon mass, a loss of OM, and an increase with sand content. Also, how could it be that the A horizons are not also disturbed by the human activity but the underlying B horizons are? Are you trying to suggest this is a plow pan? This should also be moved to the discussion section.

*Response*: Thanks very much for your elaboration of the difference in bulk density between A and B horizons. We have incorporated your suggestions and improved the explanation.

*Q22:* Line 188: When you say significantly higher, did you do a statistical test to test this? Please report your p values and test statistic here. This applies to the entire results section.

*Response*: Thanks very much for your suggestion. Where significance is mentioned

in the results, we have included *p* values and test statistics.

*Q23:* Line 232, Fig 5: Large easy to read text indicating which elements is being present. I also think this figure might be more informative as a table.

*Response*: Thanks for your suggestion. We have changed Fig. 5 into Table 4.

*Q24:* Line 233-238: How do you know that there were significant differences, please present some statistics to help us evaluate these differences.

*Response*: Thanks for your suggestion. We have added significant *p* values here.

*Q25:* Line 238-239: Again, please report statistics for these correlations.

*Response*: Thanks for your suggestion. We have added supplementary correlation statistics of these indicators here.

*Q26:* Line 239-246: What figure are you referencing when discussing the trends in CIA, CIW, and Na/K changing between landscape positions within the A, B, and C horizons?

*Response*: Thanks for your suggestion. We will revise Figure 6 to incorporate the trends in CIA, CIW, and Na/K changing between landscape positions within the A, B, and C horizons.

*Q27:* Line 244: What do you mean by "strongly detached"? I think that you are just saying that the B horizon is comprised of eroded and weathered material from higher landscape positions.

*Response*: Thanks for your suggestion. Yes, you are right. We have revised it according to your comment.

*Q28:* Line 251-254: What are the migration coefficients and how are they calculated? Are these tau values? Or some other indicator of element enrichment/depletion?

*Response*: Thanks for your suggestion. We have added to the definition and calculation formula for migration coefficients in the materials and methods section. The values given here are the average migration coefficients for different slope positions.

*Q29:* Line 262-267: How did you prep your samples for mineralogy? Are these oriented clay slides? What XRD instrument and scan parameters did you use. What treatments did you use to differentiate minerals?

*Response*: Thanks for your suggestion. We have described the procedure for using

XRD to analyze the mineral composition of soil in the materials and methods section.

**Q30:** Line 276: Please provide these statistical techniques in the methods section.

**Response**: Thanks for your suggestion. We have provided significance statistical techniques in the materials and methods section.

**Q31:** Lines 276-281: This should be moved to the result section.

**Response**: Thanks for your suggestion. We have moved lines 276-281 to results section.

**Q32:** Lines 283-292: How does your data provide a measure of Fe/Al oxide content/concentrations? Did you perform dithionite or oxalate extractions to assess the presence of crystalline/amorphous Fe/Al oxides? Further, this is rather general review information and does not provide a specific context for your results? Can you dig deeper into your data to provide some context for topographic influences on the mineralogy and geochemistry that you presented?

**Response**: Thanks for your suggestion. While we did not determine the exact content of Fe/Al oxides, we will be discussing the possible mechanism based on relevant literatures and existing studies.

**Q33:** Lines 296-298: This is redundant to what is reported in the results in lines 239-246. Please delete here and report statistics in the results section.

**Response**: Thanks for your suggestion. We have deleted this sentence in lines 296-298 and reported statistics in the results section.

**Q34:** Line 298-301: I'm really not sure what the sentence is saying. How are the soils at the summit and shoulder positions developed under natural conditions whereas other slope positions are not? Are you trying to say these have experienced less human influences? Please explain what you mean by 'natural hydrothermal status'. Are these soils impacted by hydrothermal (i.e., heated groundwater) processes? If so, that substantial changes some of the interpretations about your data.

**Response**: Thanks for your suggestion. What we aim to demonstrate here is that there is less human influences on the soil development at the summit and shoulder. We will make modifications as per your suggestions. We have changed 'natural hydrothermal status' to 'natural climate conditions'.

*Q35:* Line 302: Sodium is not a necessary nutrient for most plants. For this reason, we often use it as an indicator of chemical weathering and leaching because it is not biocycled. I also don't think that cycling of Ca, Mg would slow chemical weathering, it could change the distribution of Ca and Mg within soil profile, but there will always be some leaching and loss from the soil.

*Response*: Thanks for your suggestion. We have deleted this sentence.

*Q36:* Line 306: Instead of rust spots, do you mean redoximorphic features?

*Response*: Thanks for your suggestion. What we are trying to convey here is that at the footslope, due to the thicker soil layer and water retention, the reduction state is intensified, which results in the appearance of rust spots.

*Q37:* Line 308-312: I think it is a big leap to say that tillage causes greater chemical weathering. If you wanted to discuss this potential it would have been informative to include a representative catena that is largely unimpacted by humans.

*Response*: Thanks for your suggestion. Duo to favorable climatic conditions and population pressure, the toeslope, footslope, and even the backslope have been extensively converted to cultivated land. As a result, in the mountainous and hilly area of Sichuan Basin, there are widespread terrain sequences with woodland and grassland at the summit, and cultivated land at the footslope and toeslope. In our previous studies, we conducted laboratory and field experiments to model how anthropogenic activities, such as tillage, crop planting, fertilization, and land reclamation, accelerate the formation of purple rocks in the Sichuan Basin (Wei et al., 2006). Here, we discussed the effects of tillage on the soil formation process based on previous studies. Simultaneously, you have also given us a nice research direction. In the future, we can explore the soil formation characteristics of terrain sequences under various anthropogenic influences under natural conditions, which may be a good finding.

The references as follow:

Wei, C., Ni, J., Gao, M., Xie, D., and Hasegawa, S.: Anthropic pedogenesis of purple rock fragments in Sichuan Basin, China, CATENA, 68, 51–58, https://doi.org/10.1016/j.catena.2006.04.022, 2006.

*Q38:* Line 312-313: Unnecessary sentence. Please delete.

*Response*: Thanks for your suggestion. We have deleted the sentence in lines 312-

313.

*Q39:* Table 5: Move to the results please.

*Response*: Thanks for your suggestion. We have moved Table 5 to Section 3.3.

*Q40:* Section 4.2: Could you compare your study to other toposequence studies? I'm sure there are relevant examples of other subtropical monsoonal toposequences? Maybe these are even in similar parent material or different aspects to compare?

*Response*: Thanks for your nice suggestion. To provide a more in-depth discussion of our findings, we will compare other toposequences in relevant literature in Section 4.2.